# Why Does Hyperuricemia Not Necessarily Induce Gout?

**DOI:** 10.3390/biom11020280

**Published:** 2021-02-14

**Authors:** Wei-Zheng Zhang

**Affiliations:** VIDRL and The Peter Doherty Institute, 792 Elizabeth Street, Melbourne 3000, Australia; weizzhang@hotmail.com

**Keywords:** metabolism, inflammation, uric acid, hyperuricemia, monosodium urate crystal, gout

## Abstract

Hyperuricemia is a risk factor for gout. It has been well observed that a large proportion of individuals with hyperuricemia have never had a gout flare(s), while some patients with gout can have a normuricemia. This raises a puzzle of the real role of serum uric acid (SUA) in the occurrence of gout flares. As the molecule of uric acid has its dual effects in vivo with antioxidant properties as well as being an inflammatory promoter, it has been placed in a delicate position in balancing metabolisms. Gout seems to be a multifactorial metabolic disease and its pathogenesis should not rely solely on hyperuricemia or monosodium urate (MSU) crystals. This critical review aims to unfold the mechanisms of the SUA role participating in gout development. It also discusses some key elements which are prerequisites for the formation of gout in association with the current therapeutic regime. The compilation should be helpful in precisely fighting for a cure of gout clinically and pharmaceutically.

## 1. Introduction

Hyperuricemia has been defined as serum uric acid (SUA) >6.0 mg/dL in women; >7.0 mg/dL in men; >5.5 mg/dL in children and adolescents [1], and is an independent risk factor of a strong non-linear concentration-dependent to the incident of gout [2]. Genetic variants contribute largely to hyperuricemia [3] with 43 genes so far that have been identified in controlling SUA levels [4]. Gout, a common metabolic disorder with symptoms of localized inflammation, is caused by chronic and/or episodic deposition of monosodium urate (MSU) crystals in joints and soft tissues prompting a gouty attack/flare [5]. It has not been viewed just as an articular disease with its broad definition, but as a complex disease with interactive mechanisms of inflammatory and metabolic disorders displaying its symptoms beyond the local inflammatory consequence of MSU crystal deposition. Approximately up to 10% of adults have reported having gout [6], and 3.9% and 14.6% of the US population have gout and hyperuricemia, respectively [7].

It has been well observed and puzzled for a long time that not all hyperuricemia patients suffer from gout [8], only up to 36% of the patients develop gout [9,10], and not all gout patients have hyperuricemia [11,12]. Up to 76% of asymptomatic hyperuricemia patients could not find MSU crystal deposition [10]. This discrepancy has raised many clinical and scientific debates on the role of SUA in the development of gout and the efficacy of uric acid-lowering therapy (ULT) on the treatments of gout and other pathological conditions, despite the risk of gout increasing dramatically with increasing SUA levels in conjunction with additional factors [13]. The typical progression of gout could start from hyperuricemia without MSU crystal deposition, to crystal deposition without symptomatic gout, to acute gout flares and tophi, and to chronic gouty arthritis [14], but the correlations between gout and hyperuricemia are loose. Recent studies postulate that gout formation would be well beyond MSU crystal deposition with pathogenic mechanisms involving overproduction of chemotactic cytokines, cell proliferation and inflammation [15], and internalization of SUA induced pro-apoptotic and inflammatory effects [16].

Treatment of hyperuricemia in individuals without gout has been contentious partially due to the interrelationship between hyperuricemia and gout being not fully understood. Exploring such a mechanism, therefore, would be helpful in guiding the establishment of proper therapeutic regimes for both. This critical review aims to gather recently published literature for the elucidation of the interrelationship between hyperuricemia and gout in their pathogenesis, clinical evidence of the inflammation effects, MSU crystal formation, and therapeutic regimes, and attempts to explain why hyperuricemia does not necessarily induce gout.

## 2. Uric Acid and Gout

Uric acid (UA) is the end catabolic product of exogenous and endogenous purine nucleotide metabolism in humans. It exists in blood serum/plasma, cells, and tissues with steady-state conditions of its production and disposal. Its production can be found in almost all tissues, while its major disposal is via kidneys. SUA concentrations can be reflected by the intake from diet, in vivo purine metabolism, renal secretion, and intestinal degeneration [17]. In vivo, it may offer a neuroprotective advantage in the neurodegenerative Alzheimer’s disease [18], schizophrenia [19], Parkinson’s disease [20], multiple sclerosis [21], and serves as a depression biomarker [22]. SUA concentrations are linked to muscle strength and lean mass [23], although this was not shown in gastrointestinal tract cancer patients [24]. SUA may serve as a risk factor to predict poor thyroid function [25] or an indicator of malnutrition [15]. At a higher level, it activates inflammatory and oxidative mechanism action events in healthy subjects [26] and is a protective factor against the pathological decline of lung function [27] or an independent predictor for non-alcoholic fatty liver disease [28]. Abnormal SUA levels, either higher or lower, could increase the risk for mortality [29,30]. However, controversial reports have been presented: a slight increase in SUA level was an independent risk factor for all-cause and cardiovascular mortality [31,32], while another report did not find any relationship between SUA and cardiovascular disease (CVD) mortality and morbidity [33]. UA is also a potent antioxidant and an effective scavenger of singlet oxygen and free radicals [34], almost tenfold greater than other antioxidants in blood [26] or accounting for over half of the free radical scavenging activity in vivo [35]. Supplement of UA in donor blood sustains the antioxidant protection of the stored red blood cells [36]. The oxidant-antioxidant paradox of UA [37] may suggest UA could have different molecular behaviors under various pathological conditions. At the hydrophilic condition, it shows the protective effects of antioxidants [38]. Reducing SUA could decline its protective effect to radiation damage [39], and total bone mineral density [40,41] or a protective effect on bone loss in rheumatoid arthritis [42] (Figure 1).

Beyond its role in protection, over-saturated SUA together with sodium could deposit in joints, soft tissues, bones, skin, etc., as MSU crystals to form tophi and trigger gout flares with episodes of severe pain. Gout is a common and complex form of arthritis with a sudden attack(s) of pain, swelling, redness, and tenderness in the affected location(s). Tophaceous gout has been defined as classic periarticular subcutaneous tophi, disseminated intradermal tophi, an ulcerative form, and gouty panniculitis [43] and commonly appears as firm, pink nodules or fusiform swellings [44]. Without clinical intervention, tophi can become developed within affected joints and or tissues and progressively damage them. Interestingly, the prevalence of gout flares, irrespective of SUA levels, has been linked to mental disorders [45,46]. Chronic heart failure and diabetes mellitus are more strongly associated with increased MSU crystal deposition in knees and feet/ankles than gout duration [47]. As reducing SUA may not be the only way to eliminate gout flares [48], the level of SUA, in essence, should be an indicator of oxidative paradox in vivo.

## 3. Distinct Reaction and Priming Pathways of Inflammation by SUA and MSU

The major symptom of a gout flare is the MSU crystal-induced sterile inflammation with UA controversially being an intrinsic inhibitor of MSU crystal-induced tissue inflammation [49] and a direct promoter in inflammation in vivo [50]. Hyperuricemia could induce an activated status of inflammation [51] or autoinflammation [52]. It could instigate inflammation by stimulating the production of inflammatory factors such as interleukin-6, interleukin-1(beta), tumor necrosis factor-alpha and CRP [53], or enhancing reactive oxygen species [54], or initiating systemic inflammation via the nuclear factor (NF)-*κB* signaling pathway [55], or direct proinflammatory effects on macrophages [56]. Intra-articular injection of MSU-induced inflammatory arthritis could result from UA injection [57]. Although both SUA and MSU crystals can stimulate immune responses, different pathways have been demonstrated between them from epigenetic regulation, inflammasome activation to transcriptional control [58], and the immune reactions created by them are different inactivation and priming [59] through various proinflammatory pathways [60] involve in circulating monocytes and/or resident macrophages. UA released from dying cells during inflammation reactions could prime new monocytes and precipitate onto MSU crystals leading to more inflammation and tissue damage. Co-existing gout with hyperuricemia usually leads to higher systemic inflammation [61] and often with inflammatory psoriasis arthritis [62]. MSU deposition causes the major symptoms of inflammation and deterioration of the affected tissues with SUA playing a crucial role in a gout flare. This may support the notion that patients with both hyperuricemia and gout induce a high systemic inflammation with the greatest mortality risk [61]. Moreover, asymptomatic hyperuricemia patients usually display less potency of inflammation with a lower number of NKG2D^+^ (activating receptors of simulation of cytotoxicity) NK cells [63]. It could be speculated that under hyperuricemia condition the UA paradox interrelationship of the contradictory dual effects of inflammation [49,50] results from SUA “neutralizing” the inflammation produced oxygen species to protect tissue damages. It has also been discovered that some lipids from diet or alcohol consumption could directly trigger gout flares by activation of NALP-3 inflammasome through binding to toll-like receptors [64] or alteration of glucose and apolipoprotein metabolism [65], respectively. In conjunction with the aforementioned evidence, it may be postulated that although hyperuricemia could stimulate the formation of SUA crystals and prime immune responses in the promotion of gout flares, it goes along with different pathways of inflammation in contrast to that of MSU crystals (details have been reviewed on this topic [58]). Therefore, hyperuricemia may not be the only determinant of a gout flare or flares especially in the form of inflammation and can be present even without crystal formation in some patients [3]. Indeed, the differentiation in inflammation pathways may lay the foundation for elucidating the importance of eliminating any inflammation factors other than hyperuricemia during a gout flare.

## 4. Factors Affecting MSU Crystal Deposition Other Than Hyperuricemia

It has been found that only about half of those with SUA concentrations of ≥10 mg/dL developed clinically evident gout during a 15-year period [2]. Despite hyperuricemia playing a critical role in the formation of MSU crystals, other factors affecting MSU crystallization in tissues to induce gout flare are also involved including but not limited to temperature, pH, ion concentrations, proteins, and various connective tissue conditions as well as secondary nucleation formation [66]. Additionally, white blood cell count (WBC) in synovial envirofluid is also significantly associated with the formation of MSU crystals [67], even though it is still been unresolved whether WBC induces MSU crystal formation or *vice versa*. Reducing blood lipid levels with a lipid-lowering agent(s) could concurrently reduce SUA level [68], suggesting a decline of the hydrophobic environment could enhance the solubility of SUA to be easily excreted by the kidney instead of forming MSU. Furthermore, hyperlipidemia is more commonly seen in patients with gout in comparison with asymptomatic individuals with HDL-C being a protective predictor of SUA levels [69].

In spite of hyperuricemia, it has been established that pre-biological fiber damage could be a prerequisite for MSU crystal deposition [70]. In the predominated crystal-rich areas of gouty tophi, new crystals add on the already formed crystals to form secondary nucleation which puts pressure on surrounding cells and causes tissue damage [71]. Osteoarthritis could alter the cartilage surface to precede MSU crystal formation happening at collagen-rich sites of damaged and exposed tissues [72], and surgical tissue damage could also induce gout flare [73]. The altered composition of microbiomes could then contribute to gout formation possibly due to stone or crystal growth in vivo [74]. All of that evidence may suggest that the formation of MSU crystals to induce a gout flare usually has an abnormal environment caused by a pathological condition(s) in addition to hyperuricemia. 

## 5. The Causes of Hyperuricemia Irrelevant to Gout

An increase in UA concentration that exceeds the normal range might not be exclusively linked to gout flares [75] as many factors causing hyperuricemia are not relevant or significant to gout formation. Certain foods, status, or medicines could induce hyperuricemia. Under renal dis-function or cell damage, SUA could be suddenly increased by changing renal function to cause hyperuricemia [76]. Drugs such as diuretics (thiazide), anticonvulsants (valproate and phenobarbital), cyclosporine, theophylline, and pyrazinamide have been reported to increase SUA levels [77] in addition to favipiravir (an antiviral drug) [78]. Emotional stress [79], fasting [75], or dehydration [80] caused by physical activity can also increase the concentration of SUA. Although occasionally an acute gout flare may be linked with the medication(s) or condition(s), the enhanced SUA should be only temporarily sustained and a gout flare is considered unlikely to occur if a longer duration and higher dose treatment are avoided [81]. This may match with the fact that most gout patients do not know the trigger(s) of their gout flares [82]. Contrarily, a sudden reduction of SUA may also trigger a gout flare through the dissolution of MSU fallen off from tophi [83]. Allopurinol, a ULT medicine, has not shown any efficacy in the prevention of a first gout flare in patients with asymptomatic hyperuricemia [84]. Fundamentally, SUA level is associated with physical capacity and muscle strength in healthy subjects [23,24] and may only be a biological marker of non-gout conditions such as cardiovascular damage but is not a risk factor for its development [51]. Controversially, pseudo-gout, which is caused by calcium pyrophosphate deposition (CPPD), could have the same inflammatory symptoms as gout without hyperuricemia [85].

Gout can be self-resolvable with symptoms disappearing within days or weeks when hyperuricemia might still be sustained. Therefore, the symptoms might not be paralleled with a reduction in SUA level. The immunoreaction generated anti-UA antibody could also prevent or reduce inflammation conditions to release gout flare symptoms [62]. Under hyperuricemia, apoA-I elevation [86] or neutrophil-derived microvesicles [87] may play a role in the spontaneous resolution of acute gout arthritis.

The increased SUA after fasting for a long term may not induce any gout flares [75], plausibly as fasting could reduce the signaling pathway of the mammalian target of rapamycin (mTOR). The declined mTOR pathway could concomitantly inhibit cell growth, enhance autophagy and decrease activation of the NF-κB pathway as well as oxidative stress [88], thus reducing inflammatory processes and preventing a gout attack. All of those may be indicative that many elements inducing hyperuricemia could be irrelevant or insignificant to a gout flare. This may also at least partially explain the existence of asymptomatic hyperuricemia. However, longitudinal studies would be useful to understand the evolution of hyperuricemia and gout further and highlight the need for different treatment strategies [89]. 

## 6. Therapeutic Regimes for Treatments of Hyperuricemia and Gout

The traditional treatment for gout flares has been well established for the improvement of quality of life (Table 1), although treatment with ULT alone may not be optimal for patients [90]. The current recommendation is to additionally employ anti-inflammatory therapy [91] to reduce pathologic MSU deposition [92]. Commencing ULT alone during an acute gout flare has neither significant efficacy on localized pain, recurrent flares, or adverse effects [48] nor been associated with the risk of gout flare [93], or able to ameliorate gout associated diabetes incidence or reverse beta-cell apoptosis with significance [94] or improvement of kidney function [95]. Given the recent safety concerns, gradually up-titrated allopurinol remains the first-line ULT [96] together with concomitant colchicine or nonsteroidal anti-inflammatory drugs for enhancement of efficacy [97]. Adversely, treatment of hyperuricemia has raised attention besides gout, and current debates on whether asymptomatic hyperuricemia should be treated are still ongoing [2,98,99]. This may partially be due to both low and high SUA levels being associated with increased all-cause and cause-specific mortality with a U-shaped association [100,101]. Adventitious reduction of SUA with fenofibrate could not mediate the cardioprotective effect [68]. It may be legitimate that reducing SUA may not rapidly be able to eliminate the local inflammation that occurred upon a gout flare. Alternatively, targeting MSU crystals could be more useful and efficient in controlling gout flares. Furthermore, as some comorbidities such as chronic heart failure and diabetes mellitus are less influenced by SUA levels [47], UTL would not be favorable to them. In spite of UA infusion improving endothelial function [102], quickly lowering SUA could induce an acute gout flare [82], cardiovascular events [103], nerve dysfunction [104], the risk of all-cause death in hemodialysis [61], vertebral fracture [105], or heart failure in patients [106]. Additionally, it has been reported that in patients with gout SUA is usually lower during acute gouty attacks than during intercritical periods [11] possibly due to the slow systemic inflammatory response. Recent evidence demonstrated that SUA was increased 3 months after starting treatment with TNFis (TNF inhibitors) in inhibition of inflammation [107] with the pathogenesis remaining unknown. 

Traditional Chinese medicine treats gout with efficacy and the fundamental strategy seems to target specifically restorations of metabolisms and immunity as well as inhibition of inflammation and peripheral nerve sensation [108]. This may support the notion that gout is a systematic disease of metabolism, any symptom targeted therapy may not be the optimal strategy for a cure. Hopefully, more pathogenic evidence on the interrelationship between gout and hyperuricemia may facilitate a unified guideline globally. Fortunately, lifestyle intervention significantly decreases the MSU deposit burden and intensive training and supervision of patients with gout have resulted in very low numbers of patients not reaching treatment targets [109]. 

## 7. Conclusions

Hyperuricemia is a main, but not the only risk factor for gout flares. Recent publications suggest gout formation would be well beyond hyperuricemia in MSU crystal deposition with pathogenic mechanisms involving an activation of various inflammation pathways with UA and MSU, the critical conditions of secondary SUA deposition, and the degree of intact surrounding tissue(s). The differentiation may explain the existence of asymptomatic hyperuricemia with the UA nature of the anti-oxidative stress, gout symptom self-resolution, and some passive increase of SUA irrelevant to gout. Hyperuricemia could only be accompanied by the aforementioned factor(s) to instigate a gout flare (Figure 2). The pathogenesis between hyperuricemia and gout may support the current notion that ULT together with other therapy(s) are recommended in gout patients but controversial in asymptomatic hyperuricemia individuals, and facilitate a unified therapeutic guideline globally for both.

Key taking home messages: 1. The role of serum uric acid (SUA) should, above and beyond, be an exclusive indicator of a gout flare. 2. Hyperuricemia would not be the only risk factor of a gout flare due to many irrelevant elements between them. 3. Monosodium urate (MSU) deposition requires a pathology environment with damaged bio-fiber integrity other than hyperuricemia.

## Figures and Tables

**Figure 1 biomolecules-11-00280-f001:**
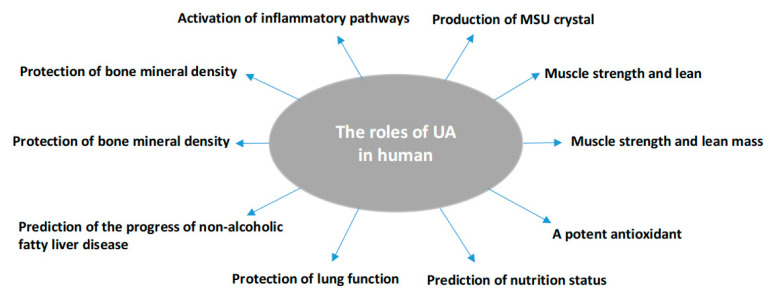
Summary of the pathophysiologic roles of uric acid in humans as identified in the review.

**Figure 2 biomolecules-11-00280-f002:**
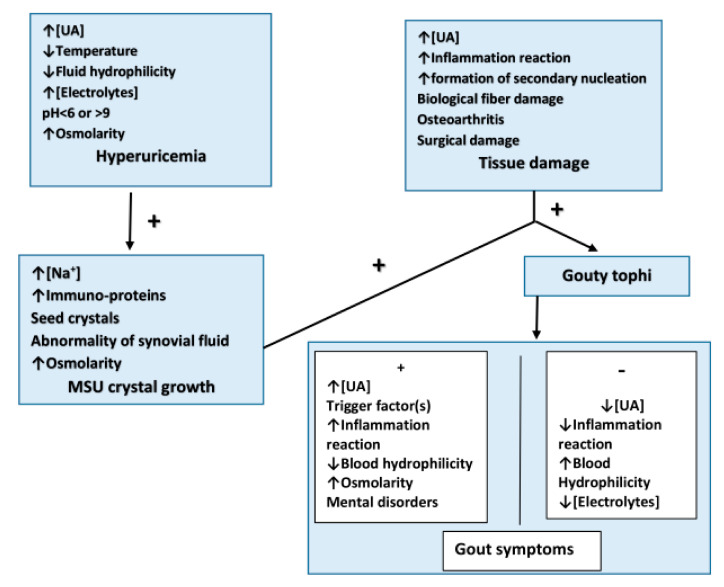
A summary of the prerequisite from hyperuricemia to gout flare. UA is involved in the pathogenesis of hyperuricemia and gout formation. Hyperuricemia could only be accompanied by the aforementioned factor(s) to instigate a gout flare. + Favorable; − Adverse; UA: uric acid.

**Table 1 biomolecules-11-00280-t001:** Common medications for the treatments of hyperuricemia and gout flares.

Drug	Brand Name(s)	Action Mechanism	Therapeutic Function
Allopurinol	Aloprim, Zyloprim	Xanthine oxidase inhibitor.	Reduces uric acid production.
Febuxostat	Ulonic	Xanthine oxidase inhibitor.	Reduces uric acid production.
Lesinurad	Zurampic	URATI inhibitor.	Helps your body get rid of uric acid when you pee.
Colchicine	Colcrys, Mitigare	Blocker of mitotic cells in metaphase.	Reduces inflammation.
Indomethacin	Indocin, Tivorbex	Inhibitor of the synthesis of prostaglandins.	Relieves the NSAID pain.
Probenecid	Probalan	Inhibition of a renal tubular transporter.	Helps the kidneys excrete uric acid from your body.
Losartan	Cozaar	Inhibition of urate/anion transport in kidneys.	Reduces uric acid levels.
Corticosteroids	Orapred, Prelone, etc.	Suppressor of the multiple inflammatory genes.	Fights with inflammation.
Fenofibrate	Antara, Fenoglide, Lipofen, Lofibra, TriCor, Triglide	Increase of uric acid solubility.	Reduces uric acid levels.
Pegloticase	Krystexxa	Recombinant urate oxidase.	Breaks down uric acid.

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
