# Peer review of "Why Does Hyperuricemia Not Necessarily Induce Gout?"

_biomolecules, 2021, doi:10.3390/biom11020280_

Round 1

Reviewer 1 Report

This manuscript summarized current knowledge about relationship between hyperuricaemia and gout. Although there are correlation between SUA and Gout, it is true that there are outliers. Many researchers feel that SUA may not be a sole cause of gout and this manuscript summarized current information on such relationship. This manuscript is well written and organized. There are only minor comments as followings:

1) Figure 2 may not be easy to wholly understand. In Figure legends, more explanation is desirable.

2)line 187 (p.7)

"deposition" is used twice. 

Author Response

Thank you very much for your review with appreciation.

In response to your constructive comments, I have my individual feed-backs below:

1) Figure 2 may not be easy to wholly understand. In Figure legends, more explanation is desirable.

I have inserted "UA involves in the pathogenesis of hyperuricemia and gout formation. Hyperuricemia could only be accompanied by the aforementioned factor(s) to instigate a gout flare." in the legend of Figure 2. 

2)line 187 (p.7) "deposition" is used twice. 

Deleted as suggested.

With kind regards

Reviewer 2 Report

1. This article has well-reviewed the roles of urate in different inflammatory pathways, and the various causes of hyperuricemia other than gout, etc.

2. ‘Gout attack/gouty attack’ can be written as ‘gout flare’, according to the article of gout nomenclature recently published for the consensus of using correct terms when referring to gout and its related complications (Gout, Hyperuricemia and Crystal-Associated Disease Network (G-CAN) consensus statement regarding labels and definitions for disease elements in gout, Ann Rheum Dis 2019;78:1592-1600).

3. It would be better to summarize the contents of part 3 (distinct reaction and priming pathways of inflammation by SUA and MSU) as a figure.

4. It would be better to list drugs according to the therapeutic function in Table 1.

Author Response

Thanks for your constructive comments. I have my responses below:

  1. This article has well-reviewed the roles of urate in different inflammatory pathways, and the various causes of hyperuricemia other than gout, etc.

Thanks again for your appreciation.

       2. ‘Gout attack/gouty attack’ can be written as ‘gout flare’, according to the article of gout nomenclature recently published for the consensus of using correct terms when referring to gout and its related complications (Gout, Hyperuricemia and Crystal-Associated Disease Network (G-CAN) consensus statement regarding labels and definitions for disease elements in gout, Ann Rheum Dis 2019;78:1592-1600).

I have read the article given and have replaced all "gout attack" with "gout flare" in the text. 

         3. It would be better to summarize the contents of part 3 (distinct reaction and priming pathways of inflammation by SUA and MSU) as a figure.

Thanks for your proactive suggestion. After re-evaluation, I may feel redundant to have a detailed figure to cover this field with the reasons below:

  1. The field has been recently reviewed exclusively with figures (Ref: 58).
  2. The mechanisms are not fully explored.
  3. This review is focused on a broad vision, and the inflammation caused by UA and MSU has already been summarised in Figure 2, adding an additional figure may overlap the contents and cause an unbalance of contents.

To compromise your suggestion, however, I have added a note of “Details have been reviewed on this topic [58]” in the text for those who want to learn more on this field.

         4. It would be better to list drugs according to the therapeutic function in Table 1.

Table 1 has been rearranged as suggested, thanks!